# An Extensive Metabolomics Workflow to Discover Cardiotoxin-Induced Molecular Perturbations in Microtissues

**DOI:** 10.3390/metabo11090644

**Published:** 2021-09-21

**Authors:** Tara J. Bowen, Andrew R. Hall, Gavin R. Lloyd, Ralf J. M. Weber, Amanda Wilson, Amy Pointon, Mark R. Viant

**Affiliations:** 1School of Biosciences, University of Birmingham, Edgbaston, Birmingham B15 2TT, UK; tjb413@student.bham.ac.uk (T.J.B.); r.j.weber@bham.ac.uk (R.J.M.W.); 2Functional and Mechanistic Safety, Clinical Pharmacology and Safety Sciences, R&D, AstraZeneca, Cambridge CB4 0WG, UK; andrew.hall@astrazeneca.com (A.R.H.); amy.pointon@astrazeneca.com (A.P.); 3Phenome Centre Birmingham, University of Birmingham, Edgbaston, Birmingham B15 2TT, UK; g.r.lloyd@bham.ac.uk; 4Clinical Pharmacology and Quantitative Pharmacology, Clinical Pharmacology and Safety Sciences, BioPharmaceuticals R&D, AstraZeneca, Cambridge CB4 0WG, UK; amanda.wilson@astrazeneca.com

**Keywords:** in vitro metabolomics, cardiac microtissues, cardiotoxicity, mode of action, biomarkers, untargeted toxicokinetics, sample harvesting, sensitivity

## Abstract

Discovering modes of action and predictive biomarkers of drug-induced structural cardiotoxicity offers the potential to improve cardiac safety assessment of lead compounds and enhance preclinical to clinical translation during drug development. Cardiac microtissues are a promising, physiologically relevant, in vitro model, each composed of ca. 500 cells. While untargeted metabolomics is capable of generating hypotheses on toxicological modes of action and discovering metabolic biomarkers, applying this technology to low-biomass microtissues in suspension is experimentally challenging. Thus, we first evaluated a filtration-based approach for harvesting microtissues and assessed the sensitivity and reproducibility of nanoelectrospray direct infusion mass spectrometry (nESI-DIMS) measurements of intracellular extracts, revealing samples consisting of 28 pooled microtissues, harvested by filtration, are suitable for profiling the intracellular metabolome and lipidome. Subsequently, an extensive workflow combining nESI-DIMS untargeted metabolomics and lipidomics of intracellular extracts with ultra-high performance liquid chromatography-mass spectrometry (UHPLC-MS/MS) analysis of spent culture medium, to profile the metabolic footprint and quantify drug exposure concentrations, was implemented. Using the synthetic drug and model cardiotoxin sunitinib, time-resolved metabolic and lipid perturbations in cardiac microtissues were investigated, providing valuable data for generating hypotheses on toxicological modes of action and identifying putative biomarkers such as disruption of purine metabolism and perturbation of polyunsaturated fatty acid levels.

## 1. Introduction

Drug-induced cardiotoxicity is a major cause of attrition during preclinical and clinical drug development and post-approval withdrawal of medicines [1,2,3,4]. Consequently, there is an urgent need to develop effective preclinical assays to evaluate the cardiac safety risk of lead compounds [2]. In vitro approaches, when coupled with molecular assays, are considered promising for this purpose given their relatively low cost, capability for high throughput and ethical benefits, i.e., in the replacement, reduction and refinement of animal research (3Rs). The effectiveness of preclinical in vitro screens is well evidenced by the successes of the hERG assay, which evaluates the risk that a compound will induce QT prolongation, an electrical disturbance in the heart [2,5,6,7]. Furthermore, the ability of assays employing stem cell-derived cardiomyocytes to predict the risk of cardiotoxicity has been demonstrated [8,9,10,11]. However, these in vitro models do not take into consideration the role of non-cardiomyocytes, which contribute approximately 70% of the cell mass and have important roles in cardiac physiology and functionality [12,13]. Furthermore, monolayer cultures are considered to lack physiological relevance given the absence of cell–cell and cell–extracellular matrix interactions which are important regulators of cell morphology [14,15,16]. The recent development and use of more complex in vitro models encompassing human induced pluripotent stem cell-derived cardiomyocytes (hiPSC-CMs) in 3D co-cultures alongside fibroblasts and endothelial cells, which together represent the majority of non-cardiomyocytes in the heart, has enhanced the specificity and sensitivity of assays which aim to predict the cardiac safety risk of screened compounds [12,17]. These spontaneously beating 3D co-cultures, composed of approximately 500 cells, are termed cardiac microtissues [17].

The majority of screening assays developed over recent years have focused on predicting functional cardiotoxicity, i.e., change in the mechanical function of the myocardium, while molecular biomarkers that can predict drug-induced morphological damage of the myocardium and loss of cardiac viability (structural cardiotoxicity) are still lacking [8,12,18]. As such, mechanistic insight and predictive biomarkers of structural cardiotoxicity remain a significant unmet need [2,8,12,18]. Untargeted metabolomics is a promising approach in toxicology for generating hypotheses on toxicity modes of action [19,20,21,22,23] and for discovering metabolic biomarkers [18]. Metabolomics-based in vivo studies have previously detected metabolic perturbations associated with drug-induced cardiotoxicity [24,25]. Furthermore, the technology has successively been applied to measure changes to the extracellular metabolome, termed the ‘metabolic footprint’, of hiPSC-CMs induced by cardiotoxins, with the responses of four metabolites (arachidonic acid, lactic acid, 2′-deoxycytidine and thymidine) proposed as biomarkers predictive of the cardiotoxicity potential of drugs [18]. Measurement of the metabolic footprint of an in vitro model can be achieved by conventional ultra-high performance liquid chromatography-mass spectrometry (UHPLC-MS)-based analyses of spent culture medium, which can be relatively simply and non-invasively sampled [18,26]. However, although it provides an indication of molecular disruption to the in vitro model, the metabolic footprint does not represent all intracellular responses [26,27]. Thus, profiling intracellular metabolic perturbations, in addition to the metabolic footprint in the culture medium, offers a more promising platform for gaining mechanistic insight and discovering predictive biomarkers of structural cardiotoxicity.

Profiling drug-induced intracellular metabolic perturbations, particularly of more advanced in vitro models such as microtissues, is challenging. For example, conventional metabolomics technologies such as UHPLC-MS typically require more than a million cells per sample [26], while 3D microtissue cultures are limited to just a few hundred cells [12]. Greater sensitivity may be achieved through application of a spectral-stitching nanoelectrospray ionisation direct infusion mass spectrometry (nESI-DIMS)-based approach, which offers an enhanced detection sensitivity while maintaining high reproducibility and minimising ion suppression and/or enhancement [28]. However, this approach is thus far untested on in vitro samples consisting of <5 × 10^5^ cells (HepaRG) [29]. Furthermore, harvesting cells in a manner that ensures the metabolomics analysis accurately measures the levels of intracellular physiological metabolites is challenging, particularly for suspended cultures including cardiac microtissues [30]. Optimally, sample harvesting removes the culture medium, including extracellular metabolites, while limiting leakage of intracellular metabolites [30]. Sample harvesting of suspended cell cultures is often achieved by centrifugation to first separate the cells from their culture medium before quenching metabolism. However, it can often take up to several minutes before metabolism is quenched, particularly when washing steps are included to ensure thorough removal of extracellular metabolites. This is deemed unacceptable for metabolites with relatively fast turnover rates [30]. Filtration has been proposed as an alternative approach which can allow more rapid isolation of cells from their culture medium, although this has only been tested on relatively large-scale cultures [30].

Characterising a drug’s disposition is commonplace in in vivo studies to aid the prediction of human risk from the responses of model species [31,32]. However, comparable analyses are not typically applied in in vitro toxicity evaluations, despite their potential for enhancing the interpretation of endogenous responses and subsequently aiding in vitro to in vivo extrapolation (IVIVE). We have previously developed and implemented an ‘untargeted toxicokinetics’ workflow capable of extracting information on xenobiotic disposition from UHPLC-MS untargeted metabolomics datasets, including extraction of relative extracellular levels of an exposure xenobiotic, sunitinib, in an in vitro cardiomyocyte-based model over a 24 h toxicity study [33]. Moving beyond this, absolute quantification of drug concentrations in the spent culture medium of in vitro models over the duration of a toxicity experiment would enable validation of drug exposure levels and may contribute to greater certainty in IVIVE.

The aim of this study was to develop and subsequently demonstrate an extensive metabolomics workflow for investigating drug-induced cardiotoxicity in cardiac microtissues, incorporating nESI-DIMS-based untargeted metabolomics of intracellular extracts and UHPLC-MS-based analysis of spent culture medium, both to profile the metabolic footprint and quantify drug exposure concentrations. Firstly, approaches for harvesting and washing cardiac microtissues for intracellular metabolomics were evaluated with respect to the *m*/*z* feature count, data reproducibility and effectiveness of culture medium removal. Additionally, an experimentally feasible sample biomass for intracellular nESI-DIMS metabolomics was determined by assessing the sensitivity and reproducibility of measurements across a gradient of cardiac microtissue biomasses. The third objective was to demonstrate the suitability of this workflow for assessing drug-induced cardiotoxicity in microtissues. Cardiac microtissues were exposed to sunitinib, a synthetic drug and established structural cardiotoxin, at two concentrations, for up to 72 h. Time-resolved intracellular metabolic perturbations and changes to the metabolic footprint were discovered by nESI-DIMS and UHPLC-MS analysis, respectively. The final objective was to integrate quantitative measurements of the extracellular concentrations of the exposure drug, sunitinib, into the same UHPLC-MS metabolomics assay.

## 2. Results and Discussion

### 2.1. Comparison of Approaches for the Sampling of Cardiac Microtissues for Untargeted Metabolomics

A rapid filtration-based microtissue sampling approach using cell strainers was evaluated by comparison against a more routine, slower centrifugation-based approach, with respect to the efficiency in isolating the microtissues from the culture medium, impact on the cells (i.e., do they induce cell lysis) and overall metabolic reproducibility. These characteristics were assessed by examining the number of *m*/*z* features detected as well as their variability across biological replicates. Comparing the time scales for the two sampling approaches (outlined in Appendix A), the filtration method required ca. 60 s from collecting the microtissues from their culture plate through to the quenching of metabolism, while the centrifugation method required over 10 min, opening the potential for significant and unwanted metabolic changes. Principal component analysis (PCA) of nESI-DIMS measurements of the metabolome and lipidome revealed that somewhat more consistent (less variable) intracellular metabolic and lipidomic signatures can be obtained when the microtissues are sampled rapidly via filtration (Figure 1). This finding is further evidenced by the lower median relative standard deviation (mRSD); an established metric for describing the variance of metabolic features [34], measured across filtered, vs. centrifuged, biological replicates (Table 1). Sampling microtissues by filtration resulted in a 1.9× and 1.6× higher number of *m*/*z* features in the intracellular polar metabolite and lipid extracts, respectively, compared to when microtissues were sampled by centrifugation. This improved coverage of the metabolome and lipidome is believed to be a consequence of the shorter handling times and reduced strain placed on cells during sampling, which likely minimises metabolite degradation and cell lysis prior to quenching. 

Thorough removal of residual culture medium from cells is preferable to limit contamination of intracellular extracts [30]. This is particularly important for nESI-DIMS measurements where high-concentration, highly ionisable compounds in the media (potentially including the exposure chemical) can induce ion competition in the mass spectrometer source, suppressing detection of the intracellular metabolome and lipidome [35]. The use of isotonic washing solutions, e.g., 0.9% NaCl (saline), is preferable in order to minimise metabolite leakage by osmotic shock to the cells [30]. However, residual salts in analytical samples can interfere with nESI-DIMS measurements; thus, a final water wash, rapid enough to minimise osmotic shock (<20 s), is recommended [36].

Consequently, we sought to optimise the procedures for washing harvested microtissues as part of the filtration-based sampling approach. This was achieved by evaluating the impact of the number and volume of saline wash steps on the nESI-DIMS measurement of process blanks, and thus the effectiveness of residual media removal, by comparing the total number of features detected per group and the relative intensities of features per group. Washing was shown to impact the amount of residual media removed from the cell strainers, demonstrated by the significant decrease in the number of features detected from washed strainers compared to no washing (Figure 2a) and a significant reduction in the intensities of between 71 and 117 features, depending on the washing protocol, that were detected in at least 2 of 3 replicates per group (*q* < 0.1, one-way ANOVA with Tukey’s HSD post hoc test and FDR correction; Figure 2b). Included in the features with significantly reduced intensities are constituents of cell culture media (e.g., putatively annotated lactic acid, amino acids; Appendix A). Upon examining a range of washing options, 3 × 1 mL saline followed by 1 × 1 mL water yielded the least number of detected features and, of those, the most features with significantly reduced intensities compared to no washing. It was concluded to wash with 3 × 1 mL saline, plus 1 × 1 mL water, to limit the residual culture medium contamination on the cell strainers during sampling of cardiac microtissues for metabolomics analysis.

Thus, here, we present a filtration-based approach which allows for rapid, reproducible and efficient harvesting of suspended cell cultures for untargeted metabolomics. Although only demonstrated for cardiac microtissues, we propose our approach would be appropriate for other suspended cell cultures of low biomass.

### 2.2. Sensitivity and Reproducibility of nESI-DIMS Untargeted Metabolomics Measurements Using Very Low Biomass Samples

Following the demonstration of an improved methodology for the sampling of cardiac microtissues relative to a more conventional centrifugation-based approach, we sought to characterise the sensitivity and reproducibility of nESI-DIMS untargeted metabolomics measurements across a gradient of cardiac microtissue biomasses, with the aim to determine which biomass to use in subsequent toxicology studies. Specifically, four biomasses were selected based on practical feasibility, corresponding to 28, 21, 14 and 7 pooled microtissues. 

The sensitivity of the nESI-DIMS untargeted metabolomics was assessed according to the number of features detected in ≥80% of samples (biological replicates and intra-study quality control samples (QCs) that were composed of an equivalent number of microtissues as the biological samples) after blank subtraction, at each sample biomass. As expected, the *m/**z* feature count increased with increasing sample biomass (Table 2). Specifically, there was a 1.7-fold and 1.8-fold increase in the number of features detected by the metabolomics and lipidomics assays, respectively, when samples consisting of 28 pooled microtissues were compared to just 7 microtissues. We concluded that the maximum feasible number of microtissues per sample, 28 (ca. 14,000 cells), should be employed to maximise detection of the metabolome and lipidome.

It is noteworthy that of the 367 metabolite and 914 lipid features detected consistently across all four sample biomasses, the intensities of 45% and 50% of these features, respectively, showed a significant positive correlation (Spearman’s, *p* < 0.05) with the sample biomass. This association evidences the biological origin of those features, thus demonstrating that a reduced feature metabolic and lipidomic signature could, in fact, be measured using nESI-DIMS analysis of as little as seven pooled microtissues (ca. 3500 cells).

The technical reproducibility of the nESI-DIMS measurements was assessed by mRSDs across replicates derived from intra-study QCs for each sample biomass. The mRSD values were ≤20.5% for all four microtissue biomasses (Table 2), which were deemed acceptable based on the mRSDs reported for cell-based samples of a larger biomass, e.g., technical mRSD of 14% for human immortalised K562 cells analysed by nuclear magnetic resonance spectroscopy [34]. 

Variation measured across biological replicates can originate from any step in the workflow from cell culturing through to the analytical measurement. Minimising this variation can improve statistical power in comparative analyses, e.g., in the discovery of metabolic features perturbed by chemical exposure. Our data show no clear trend in the total variation as a function of the sample biomass; however, the most reproducible peak intensities, as indicated by the lowest mRSD across biological replicates, are observed for samples composed of 28 pooled microtissues for both metabolomics and lipidomics measurements (Table 2), further supporting the selection of 28 pooled microtissues per sample for subsequent studies.

Thus, we have demonstrated that implementation of nESI-DIMS with SIM stitching enables reproducible untargeted metabolomic analysis of samples composed of as little as ca. 3500 cells. This discovery may encourage the application of untargeted metabolomics to other sample types which suffer from a limited biomass.

### 2.3. Probing Molecular Responses of Cardiac Microtissues to Sunitinib Exposure

With validation of a filtration-based sampling approach for cardiac microtissues and confirmation that the nESI-DIMS analytical method is sufficiently sensitive and reproducible for untargeted analyses of intracellular extracts from 28 pooled microtissues, we next sought to demonstrate the capability of this approach to discover time-resolved intracellular metabolic responses of cardiac microtissues following exposure to sunitinib, a synthetic drug and clinically labelled cardiotoxin. Additionally, we implemented UHPLC-MS(/MS) analysis of the spent culture medium from the same samples both to characterise changes to the metabolic footprint of cardiac microtissues following exposure to sunitinib and to quantify extracellular concentrations of sunitinib over the duration of exposure. 

The responses to two concentrations of sunitinib were investigated: 3.5 μM (‘high’) and 1.1 μM (‘low’), where the high concentration is phenotypically anchored to the IC_30_ of ATP depletion after 72 h of exposure [12], i.e., a concentration known to induce some metabolic disruption, and the low concentration is a half-log dilution of the high concentration.

#### 2.3.1. Phenotypic Measurements Demonstrate Adversity at 72 h

High-content biology measurements [12] conducted on a subset of microtissues provided confirmation that phenotypic perturbations are induced by both concentrations of sunitinib after 72 h. Specifically, the high exposure to sunitinib (3.5 μM) induced an 83.9% and 24.8% reduction in endoplasmic reticulum (ER) integrity and mitochondrial membrane permeability (Δψm), respectively (Appendix A). Meanwhile, the low exposure (1.1 μM) induced a slight change (−9.5%) to Δψm and a more significant 63.1% loss of ER integrity by 72 h (Appendix A). These results are consistent with previously reported dose–response relationships of sunitinib-exposed cardiac microtissues, thus demonstrating the high reproducibility of the in vitro model [12].

#### 2.3.2. Quantification of Extracellular Sunitinib in Spent Culture Medium Indicates Uptake into Microtissues

Targeted absolute quantification of sunitinib as part of the same UHPLC-MS(/MS) analytical run applied for untargeted metabolomics (described later), using an external sunitinib calibration curve (Appendix A), allowed us to confirm and evaluate the variability of sunitinib exposure concentrations across biological replicates, and to estimate the extent of uptake into the microtissues. The measurements showed extracellular sunitinib concentrations reached an equilibrium of 77% and 69% of the nominal low and high concentrations, respectively, by 2 h and remained consistent at least up to 48 h (Figure 3). This suggests a 23% and 31% uptake of sunitinib at low and high exposures, respectively, into the cardiac microtissues, assuming the nominal exposure concentrations were accurate and negligible nonspecific binding of sunitinib occurred, e.g., to plasticware. This second assumption is substantiated by there being no significant difference between the nominal and measured concentrations of sunitinib in ‘negative control’ samples that include media and sunitinib but lack microtissues. At 72 h, extracellular levels were restored to 132% and 90% of the nominal high and low exposures, respectively, a significant increase from the levels measured at 48 h (*p* < 0.0001, two-sided *t*-test), suggesting efflux of the drug from microtissues may occur between 48 and 72 h (Figure 3). The variability in exposure concentrations between samples was low, with mRSDs across biological replicates (sum of biological and technical variability) of <20% for both exposure concentrations at all time points.

Absolute quantification of exposure drugs in in vitro studies is not routinely obtained, and while measurements of an exposure drug in the same biological samples as those used to probe endogenous metabolic responses have been demonstrated previously [33], these were limited to relative exposure levels only. Here, we demonstrate that by including an external calibration curve in the analytical sequence, it is possible to absolutely quantify exposure drug concentrations using the same UHPLC-MS(/MS) untargeted measurements used to probe endogenous responses. In in vitro toxicology studies, such measurements offer the opportunity to enhance IVIVE by providing confidence in actual exposure concentrations and to make evidence-based predictions on the amount of drug uptake. 

#### 2.3.3. Untargeted Metabolomics and Lipidomics Reveal Time-Dependent Sunitinib-Induced Intracellular and Extracellular Perturbations in Cardiac Microtissues 

Four nESI-DIMS assays were implemented to measure the intracellular polar metabolome and lipidome, both in positive and negative ion modes, of cardiac microtissues exposed to sunitinib (high concentration—3.5 μM; low concentration—1.1 μM) or 0.1% (*v*/*v*) DMSO (control), sampled at 2, 6, 48 and 72 h after initial exposure. Intra-study QCs were derived from a single pool of control cardiac microtissue extracts. Biological feature counts, after data processing, were 1874 and 1559, and 2106 and 2456, for positive and negative ion metabolomics, and positive and negative ion lipidomics assays, respectively. The technical variability of the datasets was estimated from QCs, with mRSDs of 33.8% and 6.8%, and 9.0% and 14.0%, for positive and negative ion metabolomics, and positive and negative ion lipidomics assays, respectively, indicating high technical reproducibility in three of the four assays. The positive ion metabolomics dataset was deemed of poor technical quality and thus not considered further. The total metabolic variability was assessed for each group of biological replicates, with mRSDs ranging between 20.0% and 49.6% for all groups and assays (Appendix A). No clear trend in the total metabolic variability with respect to treatment class or time point was observed.

Univariate statistical analysis (one-way ANOVAs with Tukey’s HSD post hoc testing) comparing measured levels of intracellular features in sunitinib-exposed vs. control microtissues revealed a significant perturbation of the cardiac microtissue metabolome after 6, 48 and 72 h (Figure 4a). The greatest perturbation of the intracellular metabolome was observed in response to low sunitinib exposure after 6 h, where the intensities of 315 features changed significantly (*q*-value < 0.1, one-way ANOVA with Tukey’s HSD post hoc testing) compared to time-matched controls. At 48 h, the levels of 69 and 47 polar features were significantly different in microtissues exposed to low and high concentrations of sunitinib, respectively, compared to controls. Meanwhile, at 72 h, significant differences were only detected in microtissues exposed to the higher concentration of sunitinib. 

The largest response of the intracellular lipidome was observed at 48 h following high exposure to sunitinib. Here, the intensities of 100 features were significantly different compared to the control. A significant lipidomic response to the lower concentration of sunitinib was not apparent until 72 h, when the intensities of 29 features were significantly perturbed compared to the control. In contrast, the intensities of 86 features were significantly perturbed in microtissues exposed to the higher concentration of sunitinib at 72 h (Figure 4b). Further analysis, by k-means cluster analysis, revealed five subclusters of intracellular lipids displaying distinct temporal responses when exposed to the higher concentration of sunitinib. A subset of lipids in three of the five subclusters showed dose dependency in their response, with comparable temporal trends of lower magnitude observed in response to the lower concentration of sunitinib (Appendix A).

In addition to nESI-DIMS untargeted metabolomics and lipidomics of the intracellular extracts, UHPLC-MS(/MS) untargeted metabolomics was conducted on spent culture medium from the same cardiac microtissues. Feature counts after data processing and peak matrix filtering were 756 and 548 for the positive and negative ion modes, respectively. Univariate statistical analyses (one-way ANOVA with Tukey’s HSD post hoc test) revealed significant differences (*q* < 0.1) in the intensities of 18 and 28 features following exposure to low- and high-concentration sunitinib, at 6 h. Meanwhile, only 1, 2 and 1 feature(s) significantly responded to low-sunitinib exposure at 2, 48 and 72 h, respectively, and 0, 2 and 1 feature(s) significantly responded in response to high-sunitinib exposure at 2, 48 and 72 h, respectively (Figure 4c). The time of this maximal effect on the extracellular footprint is consistent with the largest response of the intracellular polar metabolome.

We further sought to discover intracellular metabolic perturbations in cardiac microtissues induced by exposure to sunitinib using supervised multivariate analyses. Partial least squares discriminatory analysis (PLS-DA) was conducted separately for each treated (either high or low concentration of sunitinib) vs. control comparison across all assays and time points. Acceptable models were built which separated the low-exposure negative ion metabolome at 2, 6 and 48 h, and high-exposure negative ion lipidome at 48 and 72 h, from the time-matched controls (Appendix A). 

Comparison of features important to the models (variable importance in projection (VIP) score > 1) at each time point revealed a relatively high level of consistency, suggesting the same set of metabolites and lipids is capable of discriminating a sunitinib-perturbed intracellular microtissue metabolome or lipidome from that of controls at any time point (Figure 5). Specifically, 191 and 388 features were consistently important in discriminating the effects of sunitinib on the metabolome and lipidome, respectively. 

Annotation of biological features can allow hypotheses to be generated with respect to the perturbations induced at the pathway level. Here, we observed a statistical enrichment (FDR-corrected *p*-value, hypergeometric test) of polar metabolites associated with purine metabolism, in response to sunitinib exposure for 6 h, as determined by univariate analyses (Appendix A).

Additionally, the intensities of three metabolites—hypoxanthine, inosine and L-glutamine (MSI level 1 annotation, using accurate *m*/*z*, retention time and MS/MS fragmentation match to reference standards, Appendix A)—were discovered to be significantly different in the spent medium of sunitinib-exposed microtissues relative to untreated controls, at 6 h (Figure 6). These metabolites are all constituents of the purine metabolism pathway (map00230, KEGG PATHWAY [37]), thereby supporting the hypothesis from the intracellular metabolomics that sunitinib perturbs purine metabolism in cardiac microtissues within 6 h of initial exposure.

Disruption of purine metabolism has also been linked with doxorubicin-induced cardiotoxicity in vitro. Specifically, an enrichment of purine metabolism pathway (hsa00230, KEGG) constituents was discovered amongst significantly down-regulated genes in doxorubicin-treated hiPSC-CMs [38]. Thus, our results, coupled to previous findings, implicate purine metabolism dysfunction in the progression of drug-induced structural cardiotoxicity. 

Furthermore, putative annotation of the features with a VIP score of >1 across multiple PLS-DA models revealed a notional importance for the response of polyunsaturated fatty acids, amongst other metabolites and lipids, in discriminating sunitinib-perturbed microtissues from controls (Appendix A). This poses these molecules as of interest for further characterisation as potential biomarkers of drug-induced structural cardiotoxicity. Polyunsaturated fatty acids have previously been shown to impact the development of drug-induced structural cardiotoxicity. Specifically, pre-treatment with eicosapentaenoic or docosahexaenoic acid significantly attenuated activation of the NF-ϰB/iNOS/NO signalling pathway, increased reactive oxygen species, change to mitochondrial membrane potential, cytotoxicity and inflammation induced by doxorubicin, a structural cardiotoxin, in H9C2 cells [39,40]. 

These results demonstrate the capability of our workflow, involving nESI-DIMS untargeted metabolomics and lipidomics of intracellular extracts, and UHPLC-MS(/MS) untargeted metabolomics of spent culture medium, to reveal metabolic and lipidomic perturbations in cardiac microtissues induced by a structural cardiotoxin, in this case, sunitinib. The time dependency of these effects reveals the importance of incorporating temporal resolution into the design of toxicity studies. Furthermore, through annotation of perturbed biological features, it is possible to develop evidence-based hypotheses on the mode of action and discover putative metabolic and/or lipid biomarkers of drug-induced structural cardiotoxicity in cardiac microtissues.

## 3. Materials and Methods

### 3.1. Cell Culturing and Sunitinib Exposure

Cardiac microtissues were cultured as described previously [12]. Briefly, primary human cardiac microvascular endothelial cells (hCMECs, Lonza, Basel, Switzerland) and primary human cardiac fibroblasts (hCFs, Lonza) were grown and maintained in EGM-2 MV Endothelial Med BulletKit (CC-3202), and FGMTM-3 Cardiac Fibroblast Growth Medium-3 BulletKit (CC-4526), respectively. On the day of microtissue synthesis, iPS cardiomyocytes (iCell Cardiomyocytes, Cellular Dynamics International (CDI), Madison, WI, USA) were thawed in plating media according to manufacturer’s instructions, and hCMECs and hCFs were detatched to obtain a single suspension. Microtissues were formed by combining cell suspensions of hiPS-CMs, hCFs and hCMECs to obtain 285 iPS-CMs, 142 hCFs and 71 hCMECs in a total of 40 uL (20 µL iPS CM thaw medium (CDI), and 20 µL EGM-2 MV medium (Lonza)) per well. Microtissues were plated into 384 ultra-low attachment U-bottom plates (Corning, UK, PN: 3830), and the medium was topped up to 80 µL with maintenance medium (50% iCell maintenance medium (CDI), 50% EGM-2 MV (Lonza)) after 48 h. Note, microtissues were not plated on the outer edge of the 384-well plates due to the higher medium evaporation rates here. Microtissues were maintained for a minimum of 14 days prior to experimentation by removal of 40 µL medium, which was replaced by fresh maintenance medium twice weekly.

Microtissues were treated with either 0.1% DMSO (control), 1.1 µM sunitinib (low concentration) or 3.3 µM sunitinib (high concentration) by the removal of 40 µL medium, and addition of drugs at a 2× stock. Microtissues were treated for either 2, 6, 48 or 72 h prior to sampling/imaging.

### 3.2. High-Content Biology Assay

Microtissues were stained for 30 min at 37 °C with fluorescent probes by removal and replacement of 40 µL cell culture maintenance medium containing 2× dyes—ER-Tracker (2 µM final) and TMRE (0.5 µM final) for ER integrity and mitochondrial membrane potential analysis, respectively. Microtissues were imaged live on a Cell Voyager 7000 (Yokogawa, Japan) using a 20× objective in a temperature (37 °C)- and CO_2_ (5%)-controlled chamber. ER-Tracker was imaged using a 405 nm excitation laser (405 ± 5 nm, 100 mW, Coherent, UK) and an Andor Neo sCMOS camera with a 445/45 nm band pass emission filter. TMRE was imaged using a 561 nm excitation laser (561 ± 2 nm, 200 mW, Coherent) and a 600/37 nm band pass emission filter. Transmitted light images were acquired using a 100 W halogen lamp as an illumination source. Images were captured over a 60 μm range in the Z-axis with a 5 μm interval between slices. Z-stack images were output as a maximal projection of multiple z-planes.

Maximum projection images were imported and analysed in the Columbus Platform (v2.7, Perkin Elmer Inc., Beaconsfield, UK). Objects were identified using the transmitted light image and the object of interest selected based on morphological and intensity-based features from the fluorescence channels. Quantitative measurements of morphological, texture and intensity features from the microtissues were captured for all parameters and exported for data analysis.

### 3.3. Microtissue Sample Collection

#### 3.3.1. Microtissue Sampling by Centrifugation

Microtissues and their culture media were aspirated from either 14 wells (1 column) or 154 wells (11 columns, corresponding to ½ plate) of the 384-well culture plate and pooled. Samples were centrifuged at 400× *g* for 3 min, the supernatant (culture medium) was discarded and the pellet (microtissues) was resuspended in 0.5 mL ice-cold 0.9% NaCl. Washed samples were centrifuged again (5000× *g*, 3 min). The supernatant was discarded, and the saline wash step repeated. Finally, the pellet was resuspended in 0.5 mL ice-cold ultra-pure water. Resuspended samples were spun until 5000× *g* was reached, and then the supernatant was rapidly discarded before quenching the microtissues in dry ice/ethanol. Harvested microtissue samples were stored at −80 °C until extraction. Process blanks were generated by sampling of media only (no microtissues) by the same approach.

#### 3.3.2. Microtissue Sampling by Filtration

We utilised an adaptation of the filtration method reported by Bordag et al. [30]. Microtissues and their culture media were aspirated from either 7 (½ column), 14 (1 column), 21 (1½ columns) or 28 (2 columns) neighbouring wells of a 384-well culture plate and added to one side of a cell strainer (37 μm reversible strainer, StemCell Technologies, Cambridge, UK). The culture medium was allowed to drain, before successive ice-cold 0.9% NaCl and ultra-pure water washes (either 2 × 0.5 mL 0.9% NaCl, 1 × 0.5 mL H_2_O for the experiments described in Section 2.1 and Section 2.2, or 3 × 1 mL 0.9% NaCl, 1 × 1 mL H_2_O for the experiments described in Section 2.3) were conducted, with the cell strainer allowed to drain. The microtissue-containing cell strainer was then flash frozen in dry ice/ethanol and stored on dry ice during the collection of remaining samples. Harvested microtissue samples were stored at −80 °C until extraction. Samples of spent culture media were spun to remove cellular debris and then stored at −80 °C until extraction.

Process blanks were generated by sampling of media only (no microtissues) using the same approach. To evaluate the effect of washing on the amount of residual medium remaining on the cell strainers after sampling, process blanks were generated using media only (no microtissues). Specifically, 2.24 mL culture medium was added to a cell strainer and allowed to drain. The strainer was then either not washed, or washed with 2 × 0.5 mL saline, 2 × 1 mL saline, 3 × 1 mL saline or 4 × 1 mL saline. Each washing protocol was concluded with a single water wash of equivalent volume to a single saline wash.

### 3.4. Intracellular nESI-DIMS Untargeted Metabolomics and Lipidomics

#### 3.4.1. Extraction of Intracellular Metabolites 

Intracellular polar metabolites were extracted from harvested microtissues by either addition of 200 μL ice-cold 4:1 (*v*/*v*) MeOH/H_2_O (LC-MS grade, VWR International, Lutterworth, UK) to each sample in a microcentrifuge tube or, for samples harvested by filtration, addition of 200 μL ice-cold 4:1 (*v*/*v*) MeOH/H_2_O to the surface of the cell strainer (held in a microcentrifuge tube, creating an airtight seal which prevents the solvent from passing through the filter), followed by re-aspiration of the solvent and microtissues, and transferred to a microcentrifuge tube. For the experiment described in Section 2.3, MeOH was supplemented with 0.2 μM L-tryptophan-(indole-d5) (Merck, Gillingham, UK), added as an internal standard to assess technical error resulting from sample preparation and analysis. Samples were vortexed for 2 min and then centrifuged (20,000× *g*, 4 °C) for 20 min. Either 180 μL of the supernatant was transferred to a microcentrifuge tube (experiments discussed in Section 2.1) or 90 μL of the supernatant was transferred to 2 microcentrifuge tubes (experiments described in Section 2.2 and Section 2.3). Extracts were then dried in a SPD11V SpeedVac sample concentrator (Thermo Scientific, Rugby, UK) for 4 h. Extraction blanks were generated by applying the same extraction procedure to fresh cell strainers.

#### 3.4.2. Extraction of Intracellular Lipids

Intracellular lipids were extracted from samples by either addition of 160 μL ice-cold MeOH to each sample in a microcentrifuge tube or, for samples harvested by filtration, addition of 160 μL ice-cold MeOH to the surface of the cell strainer (held in a microcentrifuge tube, creating an airtight seal which prevents the solvent from passing through the filter). The solvent and microtissues in the microcentrifuge tube or cell strainer were then aspirated and transferred to a 1.75 mL glass vial. An amount of 80 μL ice-cold CHCl_3_ (HPLC grade, Merck) was added, for a final solvent ratio of 2:1 (*v*/*v*) MeOH/CHCl_3_. For the experiment described in Section 2.3, MeOH was supplemented with 0.2 μM dodecylphosphorylcholine-d38 (Merck), added as an internal standard to assess technical error resulting from sample preparation and analysis. Samples were vortexed for 2 min and then centrifuged (1500× *g*, 4 °C) for 20 min. Either 200 μL of the supernatant was transferred to a 1.75 mL glass vial (experiments discussed in Section 2.1) or 100 μL of the supernatant was transferred to 200 μL conical glass inserts (experiments described in Section 2.2 and Section 2.3). Extracts were then dried by nitrogen blowdown for 10 min. Dried extracts were stored at −80 °C prior to analysis.

#### 3.4.3. Resuspension of Dried Extracts

Dried intracellular polar metabolite extracts were resuspended in either 25 µL 4:1 (*v*/*v*) MeOH/H_2_O containing 0.25% (*v*/*v*) formic acid (98%, Honeywell) or 25 µL 4:1 (*v*/*v*) MeOH/25 mM aqueous ammonium acetate (≥99.9% trace metal basis, Honeywell), for positive and negative ion metabolomics, respectively, and then centrifuged (20,000× *g*, 4 °C) for 10 min. For the experiment described in Appendix A, dried extracts were resuspended in 20 μL 4:1 (*v*/*v*) MeOH/H_2_O containing 0.25% (*v*/*v*) formic acid. Dried intracellular lipid extracts were resuspended in 35 µL 2:1 (*v*/*v*) 7.5 mM methanolic ammonium acetate/chloroform for lipidomics in positive and negative ionisation and then centrifuged (1500× *g*, 4 °C) for 10 min. 

Technical replicates for each sample class in the experiments described in Section 2.1 were generated from a pool of aliquots from the resuspended biological replicates per sampling approach. For the experiment described in Section 2.2, intra-study QCs per sample biomass were derived from a pool of aliquots from the resuspended extracts of samples with equivalent biomass. Intra-study QCs for experiments described in Section 2.3 were derived from a pool of additional biological control extracts.

Either 15 μL or 20 μL of the resuspended intracellular polar metabolite or lipid extracts, respectively, including biological samples, process blanks and intra-study QCs, were transferred to a 384-well plate (Eppendorf twin.tec PCR plate) for analysis.

#### 3.4.4. Data Acquisition by nESI-DIMS

Data were acquired using an Orbitrap Elite mass spectrometer (Thermo Fisher Scientific, Hemel Hempstead, UK) with an nESI source (TriVersa Nanomate, Advion, Ithaca, NY, USA) and the spectral-stitching nESI-DIMS method and parameters reported previously [28], with some modifications. Mainly, each sample was infused once, with each *m*/*z* window collected four times, generating ‘internal replicate’ measurements [29].

#### 3.4.5. Processing and Annotation of nESI-DIMS Data

Data were processed using DIMSpy tools [41] via the Galaxy interface as described previously [28], with some modifications. Mainly, in place of the replicate filter, only features measured in ≥75% internal replicates were retained per sample. 

Samples for which electrospray ionisation failed were excluded, based on ion injection times and total peak count (as calculated after DIMSpy peak picking, spectral stitching and internal replicate filtering). For the metabolomics (positive and negative ionisation) and lipidomics (positive ionisation only) nESI-DIMS datasets described in Section 2.3, samples were additionally filtered based on the response of the internal standard (L-tryptophan-indole-d5 and dodecylphosphorylcholine-d38 for metabolomics and lipidomics, respectively), whereby samples with an outlying internal standard peak intensity (after DIMSpy peak picking, spectral stitching and internal replicate filtering) were removed (Appendix A). The remaining samples were aligned using 3 ppm mass error tolerance for data presented in Section 2.1 and Section 2.2, and 2 ppm for data presented in Section 2.3, and data matrices of peak intensities for *m*/*z* features vs. samples were constructed. Features whose intensities in ≤80% non-blank samples were <10× their median intensity in process blanks were removed. For data presented in Section 2.3, the sample filter DIMSpy tool was also applied, removing any feature present in ≤80% across all biological samples and intra-study QCs. 

Further data pre-processing was carried out using the R/Bioconductor package structToolbox [42]. Sample filters of 75% and 80% were applied across biological and technical replicates/intra-study QCs per sample class (either sample biomass or sampling methodology) for data presented in Section 2.1 and Section 2.2, respectively, generating separate peak matrices for each sample class. For data presented in Section 2.3, samples with >30% missing values were removed from the peak matrix, and data were corrected for signal drift by quality control-robust spline correction (QC-RSC) [43]. All peak matrices were probabilistic quotient normalised (PQN), using the mean intra-study QCs as the reference. For data presented in Figure 2, the peak matrix generated by the DIMSpy ‘Align Samples’ tool was filtered so that only features present in 100% of the ‘no washing’ samples were retained. No further data pre-processing was applied.

Putative annotation of *m*/*z* features was achieved by application of the Python package BEAMSpy (Birmingham mEtabolite Annotation for Mass Spectrometry, v0.1.0, available at https://github.com/computational-metabolomics/beamspy, accessed on 4 July 2019), by accurate mass matching to HMDB (Human Metabolome Database) [44], KEGG (Kyoto Encyclopedia of Genes and Genomes) [37] and Lipid Maps [45] metabolite and lipid databases, using a 5 ppm mass error window.

### 3.5. UHPLC-MS(/MS) Untargeted Metabolomics and Quantification of Sunitinib

#### 3.5.1. Extraction of Polar Metabolites from Culture Medium

Samples of spent culture medium were prepared as described previously [46], with some minor changes. Samples were thawed on ice and briefly vortexed (5 s). An amount of 50 μL of medium was mixed with 150 μL ice-cold 1:1 (*v*/*v*) acetonitrile/MeOH (LC-MS grade, VWR international). Samples were vortexed and centrifuged (20,000× *g*, 4 °C, 20 min), and 100 μL was transferred to an HPLC vial (Chromatography Direct, UK) for analysis. Process blanks were generated from 50 μL H_2_O, in place of media. Intra-study QC samples were created by pooling an aliquot of each sample, vortexing (30 s) and then splitting into several 50 μL aliquots. Each aliquot was prepared as for the samples.

#### 3.5.2. Preparation of Sunitinib Calibration Standards, Toxicokinetic QCs and Blanks

Duplicate sunitinib calibration standards with final concentrations of 0.0625, 0.125, 0.25, 0.5, 1, 2, 4 and 8 μM were prepared in fresh microtissue culture media containing 0.1% DMSO from an initial stock solution of 50 mM sunitinib in DMSO (Honeywell). Duplicate sunitinib calibration quality control samples (‘toxicokinetic (TK) QCs’) with final nominal concentrations of 0.1875, 0.75 and 6.25 μM (high, mid and low TK QCs, respectively) were also prepared in media containing 0.1% DMSO. Triplicate ‘TK blanks’ were prepared as 0.1% DMSO in media. Then, 50 μL aliquots of the calibration standards, TK QCs and TK blanks were prepared for analysis as described in Section 3.5.1.

#### 3.5.3. Data Acquisition by UHPLC-MS(/MS)

Samples were analysed using an existing method [46] implemented on an Orbitrap ID-X Tribrid mass spectrometer (Thermo Fisher Scientific) coupled to a Vanquish Horizon UHPLC (Thermo Fisher Scientific), using an Accucore 150 Amide column (100 × 2.1 mm, 2.6 μm, Thermo Fisher Scientific) with a pre-column UHPLC filter (2.1 mm ID × 0.2 μm filter cartridge, Thermo Fisher Scientific). Mobile phase A was 95% acetonitrile/water (10 mM ammonium formate, 0.1% formic acid), and mobile phase B was 50% acetonitrile/water (10 mM ammonium formate, 0.1% formic acid) for the positive ionisation mode. For the negative ionisation mode, mobile phase modifiers were 10 mM ammonium acetate and 0.1% acetic acid, in place of ammonium formate and formic acid, respectively. The gradient was as follows: t = 0.0, 1% B; t = 2.1, 1% B; t = 4.1, 15% B; t = 7.1, 50% B; t = 10.1, 95% B; t = 11.0, 95% B; t = 11.5, 1% B; t = 15.0, 1% B. All changes were linear (curve = 5). The flow rate was 0.4 mL/min, and the column temperature was 35 °C. Analysis was performed in positive and negative ionisation modes separately at a resolution of 120,000, between 70 and 1050 *m*/*z*. Ion source parameters are detailed in Appendix A. The sample injection volume was 2 μL. 

MS/MS fragmentation data were collected by applying the AcquireX intelligent data acquisition workflow (‘DeepScan mode’, Thermo Fisher Scientific). Briefly, an initial inclusion list consisting of protonated or de-protonated ion forms of toxicologically relevant metabolites was built manually. This was added to by AcquireX using full scan data acquired from an injection of an intra-study QC. AcquireX also generated an exclusion list using full scan data acquired from an injection of a process blank. MS/MS data were then acquired using HCD with stepped normalised collision energies (NCEs) of 20, 40 and 100% and 40, 60 and 130% for positive and negative ionisation, respectively, from 3 iterative injections of an intra-study QC using the inclusion and exclusion lists, which were modified by AcquireX after each iterative injection. Analysis was performed at a resolution of 60,000 and 30,000 for full scan (MS^1^) and MS/MS, respectively, over a scan range of *m*/*z* 70–1050.

MS^1^, retention time and MS/MS fragmentation data of sunitinib, hypoxanthine, inosine and l-glutamine were also collected from chemical standards prepared in 1.5:1.5:1.0 acetonitrile/MeOH/H_2_O using identical methods and instrumentation, as detailed above.

#### 3.5.4. Processing and Annotation of UHPLC-MS(/MS) Untargeted Metabolomics Data

Vendor format raw data files (.RAW) were converted to mzML file format using ProteoWizard software [47]. Full scan (MS^1^) data deconvolution was performed by XCMS (v3.6.1) operated in Galaxy [48], as reported previously [46]. XCMS parameters were as follows: ppm (12), min. peak width (3), max. peak width (30), mzdiff (0.001), bw (0.25), minfrac (0.5), mzwid (0.01), orbiwarp retention time correction (negative ionisation data only). A data matrix of peak intensities for metabolite features (*m*/*z*–retention time pairs) vs. samples was constructed. 

Prior to data analysis, datasets were filtered as follows: any features whose median intensity in biological samples is <20× its median intensity in process blank samples were removed; features with RSD ≥ 30% across the intra-study QC samples were removed; samples with >50% missing values were removed; features which were missing in ≥10% QCs and/or ≥50% of all samples were removed. Data were corrected for signal drift by QC-RSC [43]. Peak matrices were then PQN normalised using the mean intra-study QC samples as the reference. These steps were executed using the R/Bioconductor package structToolbox [42]. 

Putative metabolite annotation was performed using BEAMSpy (v1.1.0) operated in Galaxy [48], using a 5 ppm mass error and 5 s retention time tolerance window for feature grouping. More robust compound annotations were generated through matching of MS/MS data to the mzCloud spectral database using Compound Discoverer 3.2 (Thermo Fisher Scientific). Annotations were graded by the HighChem HighRes algorithm and aligned to XCMS outputs using the R programming language, using 5 ppm mass error and 20 s retention time tolerance window. Further confidence in metabolite annotations was achieved by comparison of the MS^1^ accurate *m*/*z*, Rt and MS/MS spectrum of the specific adduct to data acquired for chemical standards using the same analytical method and instrumentation. Metabolite identification was confirmed using the following criteria: accurate mass error < 5 ppm, retention time tolerance ± 30 s and MS/MS spectrum similarity score (dot product cosine) > 0.9, as calculated using the R package, OrgMassSpecR [49], with a baseline threshold of 10% and *m*/*z* alignment tolerance of 0.001 Da.

#### 3.5.5. Quantification of Sunitinib

The concentration of sunitinib in spent culture medium of cardiac microtissues was quantified using TraceFinder (Thermo Fisher Scientific). Absolute quantification was performed using the peak area of the [M + H]^+^ ion form of sunitinib, with the identity of this peak confirmed by matching of accurate *m*/*z*, retention time and MS/MS fragmentation spectra to those of an analytical standard of sunitinib, measured as part of the same analytical sequence, i.e., Metabolomics Standards Initiative (MSI) Level 1 identification [50]. The response of sunitinib was calibrated against an 8-point calibration curve, using linear regression with 1/*x^2^* weighting to fit the curve (Appendix A). Accuracy of the calibration was deemed sufficient with the calculated amount of all but 1 of the 8 duplicated calibration points <20% of their theoretical amount. Sufficient accuracy and precision of the calibration were confirmed, with the calculated amount of 5 of 6 TK QCs < 0% of their theoretical amount (accuracy) and the coefficients of variation (CVs) per TK QC level <15% (precision).

### 3.6. Statistical Analysis

Supervised (PLS-DA) and unsupervised (PCA) multivariate analyses of nESI-DIMS metabolomics and lipidomics data were performed using the R/Bioconductor package structToolbox [42]. Prior to the analyses, the peak matrices were missing value imputed using the k-nearest neighbour algorithm (k = 5) and glog transformed, also using the structToolbox package [42].

Univariate data analysis (ANOVAs with Tukey’s HSD), to discover metabolic/lipid features with significantly different intensities between sample classes, and Spearman’s correlation analysis, to evaluate the association between *m*/*z* feature intensities and sample biomasses, were performed using the PQN-normalised peak matrices with the R/Bioconductor package structToolbox [42]. The *q*-values were calculated by false discovery rate correction.

ANOVAs with Tukey’s HSD, to test the significance between the number of features detected per sample class (washing protocol), and two-sided *t*-tests, to test the significance in changes to percentage response of ER stress and Δψm between sample classes, and the concentration of sunitinib between time points, were all performed using the R programming language (https://www.R-project.org).

K-means cluster analysis was executed using the R programming language (https://www.R-project.org). An elbow plot was generated to select the optimal value of k prior to execution of k-means cluster analysis using the log2 fold change for each *m*/*z* feature calculated per time point.

Pathway over-representation analysis was performed using putative annotations of significantly perturbed *m*/*z* features as input to the ‘Pathway Analysis’ tool of MetaboAnalyst [51]. 

## 4. Conclusions

A filtration-based approach for the sampling of cardiac microtissues for nESI-DIMS untargeted metabolomics was shown to minimise handling times, reduce metabolic variation and, with an optimised washing protocol, sufficiently remove media. Furthermore, the nESI-DIMS analytical method was deemed sufficiently sensitive and reproducible to measure intracellular metabolic and lipid *m*/*z* features in samples consisting of as few as 7 microtissues (ca. 3500 cells), while samples consisting of 28 pooled microtissues (ca. 14,000 cells) resulted in higher feature counts and more reproducible data and thus were selected for our subsequent toxicology study. We also demonstrated an extensive metabolomics-based workflow incorporating untargeted intracellular metabolomics and lipidomics by nESI-DIMS, and UHPLC-MS(/MS) analysis of spent culture medium for untargeted measurement of metabolic footprints and absolute quantification of the exposure drug, together capable of discovering time-resolved cardiotoxin-induced perturbations in a highly relevant in vitro model, cardiac microtissues. The observed metabolic and lipid perturbations provide sufficient evidence to hypothesise toxicological modes of action, including perturbation of purine metabolism and a role for polyunsaturated fatty acids, and may be further characterised for use as molecular biomarkers of drug-induced structural cardiotoxicity. 

## Figures and Tables

**Figure 1 metabolites-11-00644-f001:**
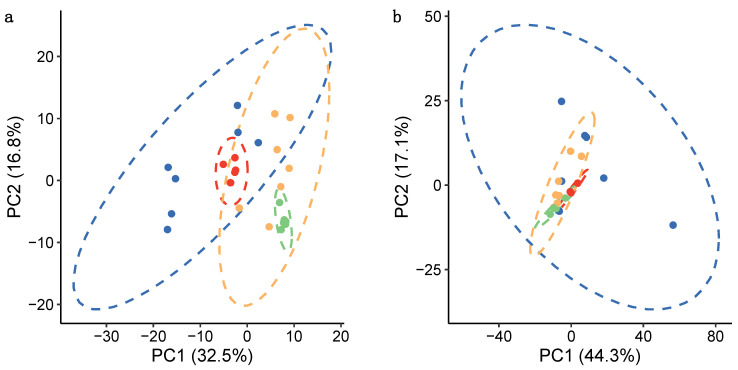
PCA score plots visualising the metabolic differences between samples harvested by centrifugation or filtration as measured by (a) nESI-DIMS polar metabolomics (positive ion mode) and (b) nESI-DIMS lipidomics (positive ion mode). Replicate samples are colour coded as: 
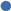
 centrifugation—biological replicates; 
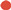
 centrifugation—technical replicates; 
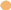
 filtration—biological replicates; and 
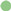
 filtration—technical replicates. Dashed lines show 95% confidence intervals for each group of samples.

**Figure 2 metabolites-11-00644-f002:**
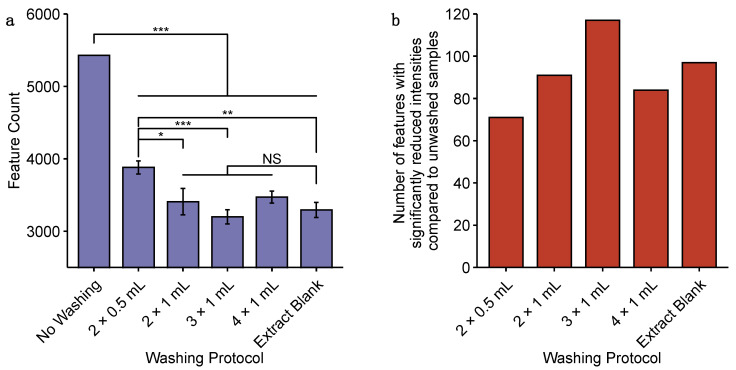
Evaluation of washing procedures for filtration-based sampling of microtissues. (**a**) Bar chart showing the total number of features detected in process blanks prepared without washing, or with 2 × 0.5 mL, 2 × 1 mL, 3 × 1 mL or 4 × 1 mL saline washes, followed by 1 × 1 mL water wash. Negative control data were acquired from extraction blanks (i.e., no medium was passed through the cell strainer prior to extraction). Error bars display standard error across 3 replicates. Significant differences between groups (one-way ANOVA with Tukey’s post hoc test) are displayed: *** *p* < 0.01, ** *p* < 0.05, * *p* < 0.1, NS, not significant. (**b**) Bar chart showing the number of features (of 2023 features detected in ≥2 of 3 replicates in each group) whose intensities are significantly reduced by each washing protocol compared to no washing. Significant reduction in feature intensity is defined where fold change of median intensity in washed samples compared to non-washed samples is <1 and *q* < 0.1, calculated by one-way ANOVA with Tukey’s post hoc test and FDR correction.

**Figure 3 metabolites-11-00644-f003:**
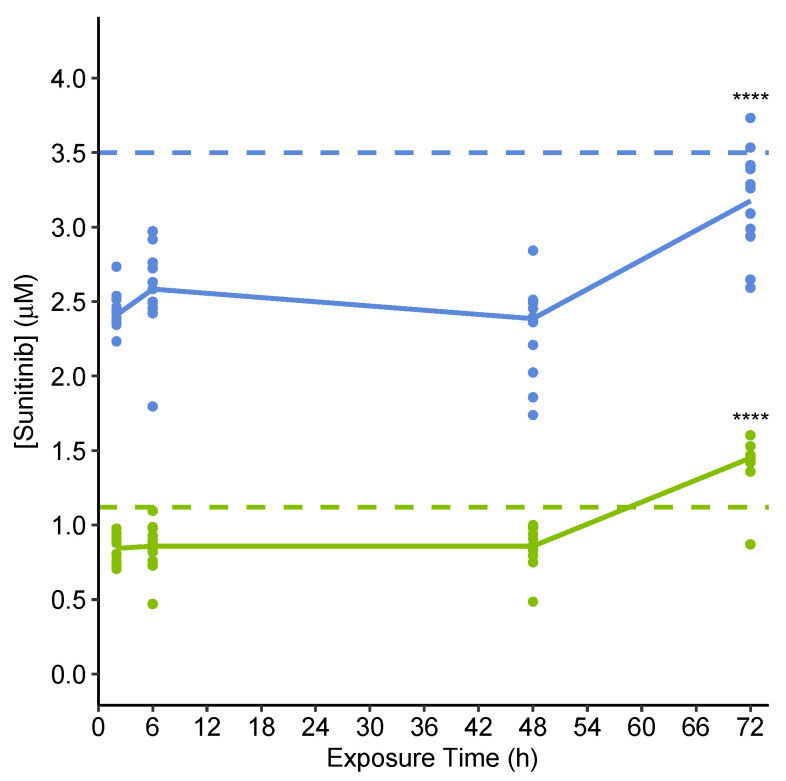
Absolute sunitinib concentrations in spent media of pooled microtissues collected at 2, 6, 48 and 72 h after initial exposure as calculated from measured peak areas using an external calibration curve. Solid lines show the median extracellular sunitinib concentration at each time point for low (green) and high (blue) exposures separately. Nominal low and high exposure concentrations are displayed as a dashed line, green and blue, respectively. **** *p* < 0.0001, for comparison of concentrations at 48 h vs. 72 h, per concentration (two-sided *t*-test). All other comparisons of concentrations at consecutive time points (2 vs. 6 h, 6 vs. 48 h) were not significant (*p* > 0.05).

**Figure 4 metabolites-11-00644-f004:**
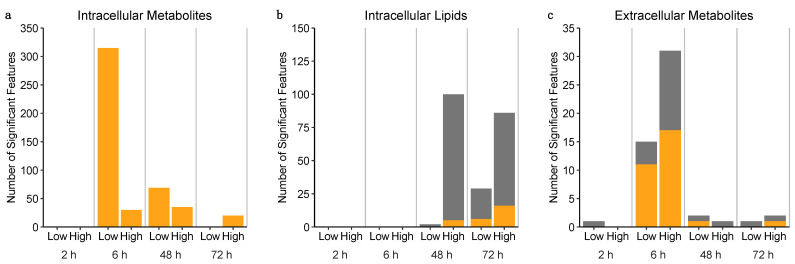
Bar charts summarising the number of significantly changing *m/z* features following the exposure of cardiac microtissues to sunitinib. Number of (**a**) intracellular metabolite, (**b**) intracellular lipid and (**c**) extracellular metabolite features whose intensities changed significantly upon exposure to two concentrations of sunitinib for 2, 6, 48 or 72 h, relative to controls. Stacked bars show contribution of measurements from positive (grey) and negative (yellow) ionisation-based assays. Significance defined where *q* < 0.1, one-way ANOVA with Tukey’s HSD post hoc test and false discovery rate correction.

**Figure 5 metabolites-11-00644-f005:**
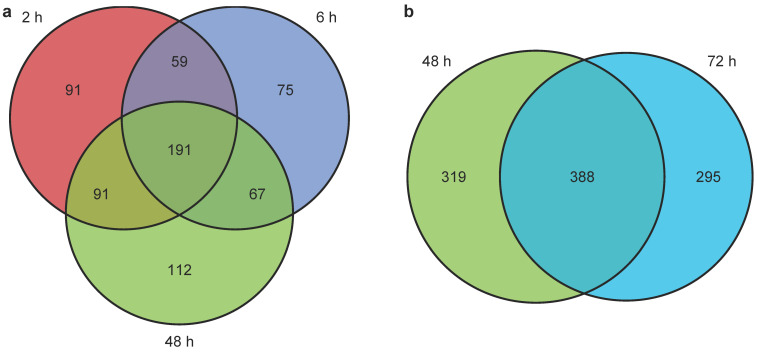
Venn diagrams displaying the overlap in features with VIP scores > 1 in PLS-DA models discriminating (**a**) between the polar metabolome, as measured by polar negative assay, of microtissues exposed to low-dose sunitinib and control microtissues at 2, 6 and 48 h after initial exposure, and (**b**) between the lipidome, as measured by lipid negative assay, of microtissues exposed to high-dose sunitinib and control microtissues at 48 and 72 h after initial exposure.

**Figure 6 metabolites-11-00644-f006:**
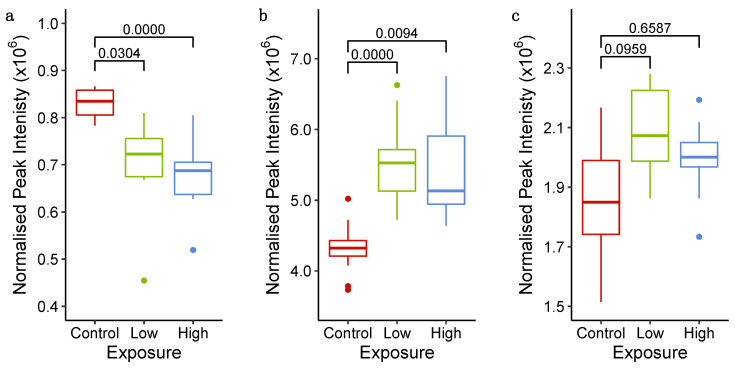
Peak intensities of (**a**) hypoxanthine ([2M − H]^−^), (**b**) inosine ([M + H]^+^) and (**c**) L-glutamine ([M – H − H_2_O]^−^) measured using HILIC UHPLC-MS in the spent culture medium of control (exposed to 0.1% DMSO), low (1.1 μM sunitinib) and high (3.5 μM sunitinib) cardiac microtissues at 6 h after exposure. The *q*-values (one-way ANOVA with Tukey’s honestly significant difference post hoc test and false discovery rate corrected) are displayed.

**Table 1 metabolites-11-00644-t001:** Metrics used to evaluate the relative coverage of the microtissue metabolome/lipidome (*m*/*z* feature count), technical reproducibility (mRSD across technical replicates) and total reproducibility (mRSD across biological replicates) for pooled microtissues sampled by either centrifugation- or filtration-based approaches and all analysed by nESI-DIMS metabolomics and lipidomics (positive ion mode).

Assay	Metabolomics		Lipidomics	
Sampling Method	Centrifugation	Filtration	Centrifugation	Filtration
Feature count	1697	3169	3173	4919
Technical variation: median RSD across technical replicates (%)	17.0	12.9	13.0	20.4
Total variation: median RSD across biological replicates (%)	42.1	35.5	50.2	31.6

**Table 2 metabolites-11-00644-t002:** Metrics to evaluate the detection sensitivity (feature count), technical reproducibility (mRSD across intra-study QCs) and total reproducibility (mRSD across biological replicates) of measurements are reported for samples of 7, 14, 21 or 28 pooled cardiac microtissues analysed by nESI-DIMS metabolomics and lipidomics (positive ion mode).

Assay	Metabolomics				Lipidomics			
Sample biomass (Number of pooled cardiac microtissues)	7	14	21	28	7	14	21	28
Feature count	796	817	1030	1321	1427	1765	2336	2507
Technical variation: median RSD across intra-study QCs (%)	13.7	10.6	18.0	20.5	17.6	16.9	15.1	12.4
Total variation: median RSD across biological replicates (%)	37.1	39.0	40.5	31.9	33.4	37.9	34.7	27.6

## Data Availability

The data presented in this study are available on request from the corresponding author. The data are not publicly available due to privacy.

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
