# Peer review of "An Extensive Metabolomics Workflow to Discover Cardiotoxin-Induced Molecular Perturbations in Microtissues"

_metabolites, 2021, doi:10.3390/metabo11090644_

Round 1
Reviewer 1 Report
The authors present a manuscript for in vitro metabolomics assessment of cardio toxins in micro tissues. The manuscript is well-written and could benefit from the following additions prior to publication:
- An expanded methods section to allow others to replicate the process exactly.
- Further discussion of perspectives from technological advancements in the future. See suggested further references below:
Leenders, J., Grootveld, M., Percival, B., Gibson, M., Casanova, F. and Wilson, P.B., 2020. Benchtop low-frequency 60 MHz NMR analysis of urine: A comparative metabolomics investigation. Metabolites, 10(4), p.155.
Huang, Q., Tan, Y., Yin, P., Ye, G., Gao, P., Lu, X., Wang, H. and Xu, G., 2013. Metabolic characterization of hepatocellular carcinoma using nontargeted tissue metabolomics. Cancer research, 73(16), pp.4992-5002. |
Hinson, J.T., Chopra, A., Lowe, A., Sheng, C.C., Gupta, R.M., Kuppusamy, R., O’Sullivan, J., Rowe, G., Wakimoto, H., Gorham, J. and Burke, M.A., 2016. Integrative analysis of PRKAG2 cardiomyopathy iPS and microtissue models identifies AMPK as a regulator of metabolism, survival, and fibrosis. Cell reports, 17(12), pp.3292-3304.
Yonezawa, K., Nishiumii, S., Kitamoto-Matsuda, J., Fujita, T., Morimoto, K., Yamashita, D., Saito, M., Otsuki, N., Irino, Y., Shinohara, M. and Yoshida, M., 2013. Serum and tissue metabolomics of head and neck cancer. Cancer genomics & proteomics, 10(5), pp.233-238.
Percival, B.C., Grootveld, M., Gibson, M., Osman, Y., Molinari, M., Jafari, F., Sahota, T., Martin, M., Casanova, F., Mather, M.L. and Edgar, M., 2019. Low-field, benchtop NMR spectroscopy as a potential tool for point-of-care diagnostics of metabolic conditions: Validation, protocols and computational models. High-throughput, 8(1), p.2.
Russell, S., Wojtkowiak, J., Neilson, A. and Gillies, R.J., 2017. Metabolic Profiling of healthy and cancerous tissues in 2D and 3D. Scientific reports, 7(1), pp.1-11.
Gonzalez-Riano, C., Garcia, A. and Barbas, C., 2016. Metabolomics studies in brain tissue: A review. Journal of Pharmaceutical and Biomedical Analysis, 130, pp.141-168. |
Percival, B., Gibson, M., Leenders, J., Wilson, P.B. and Grootveld, M., 2020. Univariate and Multivariate Statistical Approaches to the Analysis and Interpretation of NMR-based Metabolomics Datasets of Increasing Complexity.
Wu, H., Southam, A.D., Hines, A. and Viant, M.R., 2008. High-throughput tissue extraction protocol for NMR-and MS-based metabolomics. Analytical biochemistry, 372(2), pp.204-212.
Author Response
See response in red:
The authors present a manuscript for in vitro metabolomics assessment of cardio toxins in micro tissues. The manuscript is well-written and could benefit from the following additions prior to publication:
- An expanded methods section to allow others to replicate the process exactly.
We appreciate the reviewers comment and have addressed it by adding further information on the quantitation of sunitinib (Figure S3), metabolomics analytical methods (Lines 653-654, Table S7, Lines 667-670), and on metabolite annotation/identification (Lines 694-700). Besides these additions, references to other publications, where detailed descriptions of the methods can be found, have been included.
- Further discussion of perspectives from technological advancements in the future. See suggested further references below:
Some discussion of technological advancement has been added, e.g., Lines 208-212; Lines 264-268.
However we don’t understand why the reviewer has recommended many of these papers: for example, low field / low resolution benchtop NMR spectroscopy does not relate to our work. Also, cancer metabolomics studies do not relate to our work either. However, where relevant, we have added the references recommended by the reviewer to our manuscript.
Leenders, J., Grootveld, M., Percival, B., Gibson, M., Casanova, F. and Wilson, P.B., 2020. Benchtop low-frequency 60 MHz NMR analysis of urine: A comparative metabolomics investigation. Metabolites, 10(4), p.155.
Percival, B.C., Grootveld, M., Gibson, M., Osman, Y., Molinari, M., Jafari, F., Sahota, T., Martin, M., Casanova, F., Mather, M.L. and Edgar, M., 2019. Low-field, benchtop NMR spectroscopy as a potential tool for point-of-care diagnostics of metabolic conditions: Validation, protocols and computational models. High-throughput, 8(1), p.2.
Percival, B., Gibson, M., Leenders, J., Wilson, P.B. and Grootveld, M., 2020. Univariate and Multivariate Statistical Approaches to the Analysis and Interpretation of NMR-based Metabolomics Datasets of Increasing Complexity.
The above three papers refer to NMR spectroscopy and thus are not considered methodologically relevant.
Wu, H., Southam, A.D., Hines, A. and Viant, M.R., 2008. High-throughput tissue extraction protocol for NMR-and MS-based metabolomics. Analytical biochemistry, 372(2), pp.204-212.
The authors do not consider this paper (published by themselves) as appropriate to cite. Instead, more recent and relevant methodological papers have been referenced in our manuscript, e.g., Southam et al., 2018, Southan et al., 2020, Malinowska et al. (under review).
Huang, Q., Tan, Y., Yin, P., Ye, G., Gao, P., Lu, X., Wang, H. and Xu, G., 2013. Metabolic characterization of hepatocellular carcinoma using nontargeted tissue metabolomics. Cancer research, 73(16), pp.4992-5002. |
Gonzalez-Riano, C., Garcia, A. and Barbas, C., 2016. Metabolomics studies in brain tissue: A review. Journal of Pharmaceutical and Biomedical Analysis, 130, pp.141-168.
Yonezawa, K., Nishiumii, S., Kitamoto-Matsuda, J., Fujita, T., Morimoto, K., Yamashita, D., Saito, M., Otsuki, N., Irino, Y., Shinohara, M. and Yoshida, M., 2013. Serum and tissue metabolomics of head and neck cancer. Cancer genomics & proteomics, 10(5), pp.233-238.
The above three papers discuss application of metabolomics to brain or tumour tissue (i.e., in vivo), thus it is not clear how they are relevant to our manuscript.
Russell, S., Wojtkowiak, J., Neilson, A. and Gillies, R.J., 2017. Metabolic Profiling of healthy and cancerous tissues in 2D and 3D. Scientific reports, 7(1), pp.1-11.
Hinson, J.T., Chopra, A., Lowe, A., Sheng, C.C., Gupta, R.M., Kuppusamy, R., O’Sullivan, J., Rowe, G., Wakimoto, H., Gorham, J. and Burke, M.A., 2016. Integrative analysis of PRKAG2 cardiomyopathy iPS and microtissue models identifies AMPK as a regulator of metabolism, survival, and fibrosis. Cell reports, 17(12), pp.3292-3304.
The above two papers have been referenced in relation to use of 3D cultures vs. 2D.
Reviewer 2 Report
Manuscript ID: metabolites-1346014
MANUSCRIPT SUMMARY
This manuscript presents a technique to determine the minimal amount of cells to acquire metabolomic signatures on both intracellular metabolites using a nanospray direct infusion approach and the extracellular media using a more standard UPLC-MS approach. A practical experiment was also included to look at time-resolved metabolic perturbations in these cell cultures after exposure to sunitinib and also acquire quantitative measurements of the drug in extracellular media.
There is a lack of any discussion section so in the end I am left with reading 14 pages of technical details. I think in some places the amount of detail can be tightened up so it is easier to get through. Also, a majority (if not all) of the analysis is based on putative compounds.
But overall a thorough study and the authors should be commended for their attention to detail. However, quite a bit of work still needs to be accomplished.
MAJOR CONCERNS
Figure 6 and Figure S9-11, these are odd adducts for glutamine and hypoxanthine. And these are run using the HILIC and not the direct infusion. So it is imperative to include the retention time and show that a pure standard for all 3 compounds match the adducts and retention time or else the entire discussion of purine metabolism needs much more caution than simply stating in line 403 that only a MSI level 2 annotation was achieved. It should not take very long to run 3 standards.
GENERAL COMMENTS
Line 131-135, I find this paragraph either confusing or out of place. Mainly because I read 28 pooled samples in the abstract, then read 14 pooled samples to get a signal, and in line 230 I’m back to 28 pooled samples. Can this paragraph be removed or moved? Is there a purpose for its inclusion here?
Line 210-231, while I’m on this subject, as previously mentioned there is a lot of information in the manuscript and I feel that these two paragraphs are unnecessarily wordy. Your basically saying: 7 will not give you a signal, >28 is slow and impractical, so we are going to use the max number of cells to get the best signal since there is low volume. Can this be rewritten to get at this point a little quicker? Maybe the geometry of a 384-well plate is information that could go into the Figure S2 supplement or something?
Section 2.3.2, I seem to be missing what the external standard curves looked like. Do you have the r2 value to add or is it just the <20% theoretical amount listed on line 691? Was the deuterated tryptophan indole or dodecylphosphorylcholine being used for the internal standard for sunitinib quantification. This leads into line 311, is this a true absolute quantification?
Where is the discussion? Maybe this is adding to the confusion of navigating the manuscript. There is a 2. Results, then a 4. Methods with subheadings labeled with a 3.
SPECIFIC COMMENTS
Figure S2 is very helpful.
Line 259-263, very nice transition sentence.
Figure 5 may be best if moved to the supplement.
Line 403 and Figure S9-11, these can be combined into a single figure. Also, can the tandem MS please be cleaned up a bit?
Line 465, a little picky but could nM be switched to micromolar since it is used everywhere else?
Line 546, do you mean 2 mL?
Line 611, where did biofluids come from?
Line 820, this manuscript came out in 2017 correct?
Author Response
See our response below in red:
MANUSCRIPT SUMMARY
This manuscript presents a technique to determine the minimal amount of cells to acquire metabolomic signatures on both intracellular metabolites using a nanospray direct infusion approach and the extracellular media using a more standard UPLC-MS approach. A practical experiment was also included to look at time-resolved metabolic perturbations in these cell cultures after exposure to sunitinib and also acquire quantitative measurements of the drug in extracellular media.
There is a lack of any discussion section so in the end I am left with reading 14 pages of technical details. I think in some places the amount of detail can be tightened up so it is easier to get through. Also, a majority (if not all) of the analysis is based on putative compounds.
Each of these points is responded to below, where the reviewer has provided more detailed descriptions of their concerns and recommended changes.
But overall a thorough study and the authors should be commended for their attention to detail. However, quite a bit of work still needs to be accomplished.
MAJOR CONCERNS
Figure 6 and Figure S9-11, these are odd adducts for glutamine and hypoxanthine. And these are run using the HILIC and not the direct infusion. So it is imperative to include the retention time and show that a pure standard for all 3 compounds match the adducts and retention time or else the entire discussion of purine metabolism needs much more caution than simply stating in line 403 that only a MSI level 2 annotation was achieved. It should not take very long to run 3 standards.
As requested, new data showing MS1, MS/MS and retention time matches to pure standards for the adducts reported in the manuscript have been added. Metabolite standard data was collected using the same methods, on the same instrument, as we have described for the study samples (in the paper) and therefore our discussion of purine metabolism is now based on MSI Level 1 metabolite identifications. These data are presented in new Figure S9 and Figure S10. We thank the reviewer for highlighting this point, and subsequently leading to an improved manuscript.
GENERAL COMMENTS
Line 131-135, I find this paragraph either confusing or out of place. Mainly because I read 28 pooled samples in the abstract, then read 14 pooled samples to get a signal, and in line 230 I’m back to 28 pooled samples. Can this paragraph be removed or moved? Is there a purpose for its inclusion here?
The authors acknowledge the confusion described by the reviewer and have thus removed these lines from the manuscript, recognising that they added little value. Figure S1, i.e., the figure presenting the data being described in these lines, has also been removed. The flow of the text is now much improved.
Line 210-231, while I’m on this subject, as previously mentioned there is a lot of information in the manuscript and I feel that these two paragraphs are unnecessarily wordy. Your basically saying: 7 will not give you a signal, >28 is slow and impractical, so we are going to use the max number of cells to get the best signal since there is low volume. Can this be rewritten to get at this point a little quicker? Maybe the geometry of a 384-well plate is information that could go into the Figure S2 supplement or something?
The authors appreciate the reviewers comment, recognising that these lines go into a lot of detail on the selection of sample biomasses. We have tried to tighten the text here, as recommended by the reviewer. Some text regarding microtissue culture plate geometry has been added to the Materials and Methods, where the authors believe it is relevant.
Section 2.3.2, I seem to be missing what the external standard curves looked like. Do you have the r2 value to add or is it just the <20% theoretical amount listed on line 691? Was the deuterated tryptophan indole or dodecylphosphorylcholine being used for the internal standard for sunitinib quantification. This leads into line 311, is this a true absolute quantification?
The external calibration curve for sunitinib has been added to the supplementary information, see Figure S3. Included in the figure legend is the R2 value (0.99). The deuterated compounds mentioned were not used in the LC-MS analyses. The authorship team strongly believe, based on many years of experience working in pharmacokinetics (at AstraZeneca), that this is true absolute quantification through use of matrix-matched external calibration and quality control samples.
Where is the discussion? Maybe this is adding to the confusion of navigating the manuscript. There is a 2. Results, then a 4. Methods with subheadings labeled with a 3.
The authors have combined the results and discussion into a single section in this manuscript, which, according to the ‘Instructions for authors’, is accepted by the journal. We apologise for the subheading typos, they should read “2. Results and Discussion”, “3. Materials and Methods”, and “4. Conclusions”. These have been updated in the revised manuscript.
SPECIFIC COMMENTS
Figure S2 is very helpful.
Line 259-263, very nice transition sentence.
We thank the reviewer for these comments.
Figure 5 may be best if moved to the supplement.
We believe that Figure 5 shows valuable information, mainly the consistency in important metabolic and lipid features for discriminating between control and treated microtissues, as inferred from PLSDA VIP scores, between time points. We believe this visual representation, alongside the explanatory text (Lines 394-399), helps provide clarity to the reader and should therefore be included in the main manuscript.
Line 403 and Figure S9-11, these can be combined into a single figure. Also, can the tandem MS please be cleaned up a bit?
Figures S9-S11 have been cleaned up, now Figure S11.
Line 465, a little picky but could nM be switched to micromolar since it is used everywhere else?
Yes, this has been changed in the revised manuscript.
Line 546, do you mean 2 mL?
We apologise for the typo. 200 uL conical glass inserts were used. This is reflected in the revised manuscript.
Line 611, where did biofluids come from?
The authors understand the possible confusion here. We have changed to “culture medium” to avoid confusion and ensure consistency with the rest of the manuscript.
Line 820, this manuscript came out in 2017 correct?
We apologise for the typo. The reference has been corrected.
Reviewer 3 Report
This is very perfect manuscript for this journal. The authors followed the very unique concept to identify the drug-induced cardiotoxicity-based generation of potential biomarker. Which can help to understand the mis-regulation of any particular metabolic pathways. In this manuscript author chose the filtration-based approach for the sampling of cardiac microtissues for untargeted metabolomics. Their analysis supported the evidence that includes modification of purine metabolism and a role for polyunsaturated fatty acids as a potential biomarker for cardiotoxicity. Methodology description and data analysis was done well. In this manuscript I can see the authors did some biological assay like micro-tissue imaging to find the mitochondria and ER integrity. I did not see this data in this manuscript. I strongly suggest putting in this manuscript. Next, I found the many of metabolites excel sheet. Please prominent metabolite collects and put in table form. I would expect to include all my suggestion in this manuscript before acceptance.
Author Response
See our responses below in red:
This is very perfect manuscript for this journal. The authors followed the very unique concept to identify the drug-induced cardiotoxicity-based generation of potential biomarker. Which can help to understand the mis-regulation of any particular metabolic pathways. In this manuscript author chose the filtration-based approach for the sampling of cardiac microtissues for untargeted metabolomics. Their analysis supported the evidence that includes modification of purine metabolism and a role for polyunsaturated fatty acids as a potential biomarker for cardiotoxicity. Methodology description and data analysis was done well.
The authors would like to thank the reviewer for these comments.
In this manuscript I can see the authors did some biological assay like micro-tissue imaging to find the mitochondria and ER integrity. I did not see this data in this manuscript. I strongly suggest putting in this manuscript.
Please see Figure S2 for the phenotypic assay data in question.
Next, I found the many of metabolites excel sheet. Please prominent metabolite collects and put in table form. I would expect to include all my suggestion in this manuscript before acceptance.
We appreciate there is a lot of information presented in the supplementary tables (found within the Excel file), however we believe it is all valuable and it is not clear how to identify “prominent metabolites”. Presenting information from a metabolomics study in the manner we have done is commonplace in our field.
Round 2
Reviewer 2 Report
The authors did an excellent job handling the reviews.